# The Use of κ-Carrageenan in Egg Yolk Free Extender Improves the Efficiency of Canine Semen Cryopreservation

**DOI:** 10.3390/ani12010088

**Published:** 2021-12-31

**Authors:** Eunji Kim, Areeg Almubarak, Nabeel Talha, Il-Jeoung Yu, Yubyeol Jeon

**Affiliations:** 1Department of Theriogenology and Reproductive Biotechnology, College of Veterinary Medicine and Bio-Safety Research Institute, Jeonbuk National University, Iksan 54596, Korea; dmswl2570@naver.com (E.K.); areegalmubarak@yahoo.com (A.A.); iyu@jbnu.ac.kr (I.-J.Y.); 2Department of Veterinary Medicine and Surgery, College of Veterinary Medicine, Sudan University of Science and Technology, P.O. Box 204, Hilat Kuku, Khartoum North 11111, Sudan; nabeeltalha@gmail.com

**Keywords:** apoptosis, biobank, canine sperm, cryosurvival, κ-carrageenan, semen freezing

## Abstract

**Simple Summary:**

Semen cryopreservation generates sperm damage and reduces fertilization capacity as a consequence of reactive oxygen species formation, a known trigger of the apoptosis cascade. The present study aimed to examine the effects of κ-carrageenan supplementation to the freezing medium on canine sperm cryo-survival. To this end, the sperm was frozen in egg-yolk-free diluent containing different κ-carrageenan concentrations (0.0%, 0.1%, 0.2%, 0.3%, and 0.5%). The addition of κ-carrageenan to the extender at a 0.2% concentration resulted in a significant increase in the total motility and rapid progressive motility. Increasing levels of κ-carrageenan increased the spermatozoa acrosome integrity (*p* < 0.05). Apoptosis levels were significantly lower in the 0.1% and 0.2% κ-carrageenan treatment. Moreover, the κ-carrageenan supplemented group exhibited an upregulation of the relative expression of anti-apoptotic and down-regulation of oxidative state-related genes (*p* < 0.05). Overall, the introduction of κ-carrageenan as a component of egg-yolk-free freezing diluent could be useful for defining the cryopreservation media and improving the overall efficiency of cryopreserved canine semen.

**Abstract:**

κ-Carrageenan is a plant polysaccharide derived from red seaweeds reported to possess potential medicinal and antioxidants activities. The present study aimed to identify the cryoprotective effects of κ-carrageenan on the quality of frozen-thawed canine semen. Twenty-eight ejaculates were collected and diluted in a Tris egg-yolk-free extender supplemented with various concentrations of κ-carrageenan (0.0%, 0.1%, 0.2%, 0.3%, and 0.5%). The addition of κ-carrageenan to the extender at a 0.2% concentration induced a significant increase in the total motility (TM) and the rapid progressive motility (RPM) of canine sperm. Among the experimental groups, the highest percentage of sperms with intact acrosomes was found in the 0.5% κ-carrageenan group (*p* < 0.05). Apoptosis levels were significantly lower in the 0.1% and 0.2% κ-carrageenan treatment. Moreover, sperm in the κ-carrageenan supplemented group showed a significantly higher expression of antiapoptotic (Bcl-2) and lower expression of NADPH oxidase (NOX5), spermine synthase (SMS), and spermine oxidase (SMOX) genes than those in the control group. In conclusion, the addition of κ-carrageenan to the freezing extender improved the overall efficiency of frozen-thawed dog spermatozoa.

## 1. Introduction

Cryopreservation is a valuable tool for the long-term storage of semen and the distribution of genetic resources [1]. Despite numerous attempts to improve freezing media and techniques, the main issue with cryopreservation procedures is the deterioration of sperm quality. As a result of thermal, mechanical, osmotic, and oxidative stress, the quality of frozen-thawed sperm is severely compromised, and sperm undergoes some detrimental changes at the structural and molecular levels [2,3]. The sperm that survives after freezing-thawing exhibit a reduction in fertility and this has been linked with damage that adversely affects viability, motility, plasma membrane, and acrosome integrity. In addition, the activation of apoptosis pathways results in the fragmentation of sperm cell DNA. These changes ultimately contribute to an overall reduction in the fertility of sperm [4,5].

Numerous approaches have been attempted to improve the quality of frozen-thawed (FT) sperm. The addition of cryoprotectant agents such as glycerol, egg yolk, milk, polyvinyl alcohol, and cholesterol to extenders have been reported to reduce the detrimental effects of the freezing and thawing process [6,7,8,9,10,11]. Since its discovery as a component of cooling extender, egg yolk has been widely used in mammalian sperm cryopreservation to protect sperm from initial cold shock [12,13]. Moreover, many studies have verified the benefits of egg yolk-based extenders for canine semen cryopreservation [14,15]. However, it has been stated that the use of egg yolk interferes with sperm evaluation and the presence of particulate material in the extender may reduce fertility [16,17]. Aside from the risk of microbial contamination, variation in egg yolk composition due to differences in breed and management [12,18] has prompted the search for potential animal-free substitutes.

Carrageenan is a plant polysaccharide derived from Irish moss and obtained by thermal stratification from Rhodophyceae [19]. It has been reported to be used as a food additive and as a treatment for diseases such as arthritis [20]. When carrageenan is dissolved in an aqueous solution, it increases the viscosity of the solution and forms the structural component of the polymer matrix [21]. κ-Carrageenan was reported to improve rooster sperm cryo-survival through diminishing the intracellular reactive oxygen species (ROS) and the up-regulation of endogenous antioxidants enzymatic activities [22]. Furthermore, several studies indicate that plant polysaccharides have antioxidant activities and can be used as new potential antioxidants [23,24,25]. In the present study, we evaluate if treating canine sperm with κ-carrageenan prior to cryopreservation could improve the quality of FT spermatozoa. We assessed sperm motility, viability, acrosome integrity, apoptosis, and ROS levels. In addition, the expression profile of genes related to apoptosis and oxidative status were also examined.

## 2. Materials and Methods

### 2.1. Reagents and Composition of Extenders

Unless otherwise noted, all chemicals used in the present study were purchased from Sigma-Aldrich (St. Louis, MO, USA).The extenders used in the present study were extender 1 which consists of 2.42% [*v/v*] Tris, 1.34% [*v/v*] citric acid, 0.8% [*v/v*] glucose, 0.06% [*v/v*] penicillin, and 0.1% [*v/v*] streptomycin in distilled water (Invitrogen, Carlsbad, CA, USA) and various κ-carrageenan concentrations (0.0%, 0.1%, 0.2%, 0.3%, and 0.5%). Extender 2 was composed of extender 1 plus 9.2% [*v/v*] glycerol [8].

### 2.2. Animals and Semen Collection

Five sexually mature and healthy males (two mixed breeds, one poodle, one dachshund, and one jindo) aged 3–5 years were used for semen collection. All dogs were housed individually in cages furnished with all the necessary animal care facilities following the Guiding Principles for the Care and Use of Research Animals established by Jeonbuk National University (JBNU, 2019-096). Semen was collected twice a week using the gloved-hand method.

### 2.3. Experimental Design

A total of twenty-eight ejaculates were used to investigate the effects of κ-carrageenan supplementation to the freezing medium on canine sperm cryo-survival. In experiment one, the effect of adding different concentrations of κ-carrageenan during canine semen cryopreservation on FT sperm quality was assessed. The eligible ejaculates (sperm concentration > 200 × 10^6^ spermatozoa/mL, motility ≥ 80%) were pooled, extended (control, 0.1%, 0.2%, 0.3%, and 0.5% κ-carrageenan), cooled, and cryopreserved as described below. After thawing, sperm motility, progressive motility, plasma membrane, and acrosome integrity for each group were evaluated. In experiment two, the effect of various concentrations of κ-carrageenan on apoptosis and ROS levels of FT spermatozoa were assessed. In experiment three, the relative expression of selected genes related to apoptosis and oxidative status was investigated in the κ-carrageenan 0.1% treatment and control group.

### 2.4. Semen Cryopreservation and Thawing

Eligible samples were centrifuged at 300 *g* for 12 min and the pellets were pooled to overcome individual variations [26]. The pellet was then diluted to 2 × 10^8^ sperm/mL in extender 1. The extended semen was incubated at 4 °C for 1 h. Semen was further diluted (1:1) with extender 2 to achieve a sperm concentration of (1 × 10^8^ sperm/mL). The diluted semen was packed into 0.5 mL straws (FHK, Tokyo, Japan) and incubated at 4 °C for 20 min. Equilibrated straws were frozen in liquid nitrogen vapor (LN_2_) for 20 min by placing them horizontally 1–2 cm above the surface of LN_2_ in a covered polystyrene foam box (30 cm × 23 cm × 24 cm) prior to merging them into LN_2_. Thawing was performed in a water bath at 38 °C for 25 s.

### 2.5. Assessment of Frozen-Thawed Semen Motility

Frozen-thawed (FT) sperm motility was measured using computer-assisted sperm analysis (Sperm Class Analyzer, Microptic, Barcelona, Spain). The SCA settings were adjusted as described previously [27]. Briefly, 3 μL of FT semen was placed in a counting chamber (Leja, Nieuw-Vennep, Netherlands) on a warm plate at 38 °C. For each analysis, five fields were evaluated and at least 1500 spermatozoa were counted. Motility patterns including total sperm motility (TM), rapid progressive motility (RPM), medium progressive motility (MPM), and immotility (IM) were measured.

### 2.6. Assessment of Plasma Membrane Integrity

The plasma membrane integrity of FT spermatozoa was assessed using a LIVE/DEAD sperm viability kit (Molecular Probes, Eugene, OR, USA) in accordance with a previous study [28]. The assay was performed by adding 1 mM SYBR-14 dye and then incubating for 5 min in the dark. Then, 2.5 mM propidium iodide (PI) was added to the stained sperm for another 5 min. Afterward, samples were smeared on glass slides and air-dried. The smears were captured under a fluorescence microscope (Axio, Carl Zeiss Microscopy, White Plains, NY, USA). Appropriately 200 spermatozoa were counted per slide and classified as intact plasma membrane (green fluorescent) or membrane damaged spermatozoa (red fluorescent).

### 2.7. Assessment of Acrosome Integrity

The acrosome integrity was evaluated using pisum sativum agglutinin (PSA) conjugated to fluorescein isothiocyanate (FITC) staining as described previously by [29]. Briefly, FT semen was centrifuged at 400× *g* for 4 min, and the supernatant was then removed. The sperm pellet was resuspended in 500 µL of 1 × Dulbecco’s phosphate-buffered saline (DPBS, Gibco, Grand Island, NY, USA). Afterward, a 30 µL aliquot was spread on a glass slide and dried for 10 min. The slides were then incubated for 10 min, stained with 30 µL of PSA-FITC, dried, and fixed with absolute alcohol. The smear was covered with parafilm (Bemis, Chicago, IL, USA) for 20 min and then immersed in distilled water for 15 min. The dried smear was examined by fluorescence microscopy (Axio, Carl Zeiss Microscopy, White Plains, NY, USA). The percentage of acrosome-intact sperm (spermatozoa with a strong green fluorescence on the acrosomal region) was counted in a minimum of 200 sperms per slide.

### 2.8. Apoptosis Index

Apoptosis was assessed using the FITC Annexin V apoptosis detection kit I (BD Biosciences, San Diego, CA, USA) in accordance with the manufacturer’s instructions. Briefly, FT spermatozoa were centrifuged at 300× *g* for 5 min. The sperm pellet was resuspended in 1 × Annexin V binding buffer (10 mM HEPES/NaOH (pH 7.4), 140 mM NaCl, and 2.5 mM CaCl_2_). Next, 5 µL of Annexin V-FITC and 5 µL of PI were added to a 1.5 mL microtube containing sperm under dark conditions. After incubation for 15 min, 400 µL of 1 × Annexin V binding buffer was added, and the samples were then assessed using a FACSCalibur^®^ flow cytometer (Becton Dickinson, Franklin Lakes, NJ, USA). For the assessment, the labelling patterns in the Annexin (AN)/PI analysis were classified as follows: viable (AN−/PI−), viable but phosphatidyl serine (PS) translocated (AN+/PI−), dead and PS translocated (AN+/PI+), and dead and late necrotic sperm (AN−/PI+). The apoptosis index was calculated as the ratio between AN+/PI− spermatozoa and total viable (PI−) spermatozoa.

### 2.9. ROS Level

A FACSCalibur^®^ flow cytometer was used to analyze the levels of intracellular ROS of FT spermatozoa following the method described previously by [30]. In brief, FT samples were resuspended in 1 × DPBS. Afterward, 20 mM 2′, 7′-dichlorodihydrofluorescein diacetate (H_2_DCFDA DCF, Invitrogen, Carlsbad, CA, USA) and PI were used to detect intracellular free radicals and the samples were incubated for 1 h at 25 °C in the dark. Afterward, the mean fluorescence intensity (MFI) of DCF was measured to evaluate intracellular mean H_2_O_2_ per viable sperm population.

### 2.10. Gene Expression Analysis

Total RNA was extracted from FT samples in the control and the 0.1% κ-carrageenan group using the RiboEx reagent (GeneAll) according to the manufacturer’s instructions. Quantitative real-time PCR (qPCR) was conducted to evaluate transcript abundance using oligonucleotide primer sequences shown in Table 1.

Total RNA samples were adjusted for an equal starting concentration to achieve a standard efficiency among all selected genes. The mRNA expression of B-cell lymphoma 2 (Bcl-2), Bcl-2-associated X (BAX), the NADPH oxidase (NOX5), spermine oxidase (SMOX), and spermine synthase (SMS) were assessed using the qPCR assay. The qPCR assay was conducted using the One-Step TB Green^®^ PrimeScript™ RT-PCR Kit II (Takara, Shiga, Japan) and an ABI 7300 real-time PCR system (Applied Biosystems, Beverly, MA, USA). The experiments were performed using the following PCR setup: stage 1, reverse transcription: 5 min at 42 °C and then 10 s at 95 °C; stage 2, PCR reaction: 40 cycles of 5 s at 95 °C and 34 s at 56 °C; and stage 3, dissociation. The target genes were quantified relative to the expression of the internal control gene (glyceraldehyde 3-phosphate dehydrogenase, GAPDH) using the equation R = 2^−(∆Ct sample − ∆Ct control)^ as described previously [31]. For the easiness of comparison, the mean expression level of each gene from the control group was set as 1.

### 2.11. Statistical Analysis

Data were analyzed using SPSS 2.0 (IMN, New York, NY, USA) except for gene expression analysis. All experiments were conducted with a minimum of three replicates. Data related to the determination of the optimal κ-carrageenan concentration were analyzed using a one-way analysis of variance (ANOVA) and Tukey’s multiple comparison test. The results of the gene expression were assessed using the statistical analysis system ver. 8x (SAS Institute Inc., Cary, NC, USA). Data were compared by one-way ANOVA, followed by Duncan’s multiple range test. All values are presented as the mean ± SE, and a probability value of *p* < 0.05 was considered to be statistically significant.

## 3. Results

### 3.1. Effect of κ-carrageenan on Sperm Motility

The motion parameters of canine semen after freezing-thawing in extenders containing various concentrations of κ-carrageenan are presented in Table 2.

There was a significant difference regarding TM (*p* < 0.05) for the group treated with extender containing 0.2% κ-carrageenan. Additionally, the percentage of RPM spermatozoa was significantly higher in the 0.2% and 0.1% κ-carrageenan treatment than in other groups.

### 3.2. Effect of κ-carrageenan on Sperm Plasma Membrane Integrity 

As shown in Figure 1, the percentage of FT spermatozoa with an intact plasma membrane did not vary statistically among the experimental groups.

### 3.3. Effect of κ-carrageenan on Acrosome Integrity

As shown in Figure 2, the percentage of spermatozoa possessing intact acrosomes tended to be higher with an increased κ-carrageenan concentration. Significantly higher numbers of sperms with intact acrosomes were found in the group treated with 0.5% κ-carrageenan compared to those numbers in the other groups (*p* < 0.05). 

### 3.4. Effect of κ-carrageenan on Apoptosis

Figure 3 presents the apoptosis results for the canine FT spermatozoa treated with various concentrations of κ-carrageenan. Supplementation of extender with 0.1% and 0.2% κ-carrageenan significantly decreased levels of apoptosis compared to that of the other groups (*p* < 0.05).

### 3.5. Effect of κ-carrageenan on ROS Levels

The effect of different concentrations of κ-carrageenan on ROS levels of FT sperm was shown in Figure 3. In general, the ROS levels tended to be lower with high κ-carrageenan treatment. The 0.5% treatment group exhibited the lowest ROS level than other groups. However, the observed difference was not significant (*p* > 0.05).

### 3.6. Effect of κ-carrageenan on Gene Expression

The relative expression level of the antiapoptotic gene (Bcl-2) was significantly upregulated in the κ-carrageenan group compared to that of the control group. However, the expression of (BAX) did not differ statistically between the two groups, while the expression of (NOX5, SMOX, and SMS) exhibited significantly lower levels in the κ-carrageenan group than those observed in the control group (Figure 4).

## 4. Discussion

Egg yolk-based extender exerts a protective effect on FT semen [4,32]. However, particulates in yolk induce difficulty during semen evaluation and may interfere with FT semen quality. In addition to increasing the emphasis on biosecurity issues and controlling disease with international semen shipment [12,17], this all prompted the search for an extender composed entirely of chemically defined components to achieve more effective cryopreservation. In the present study, we examined the effect of various κ-carrageenan concentrations supplemented with Tris-based egg yolk-free extender on FT dog sperm quality.

The motility of sperm is crucial for biological functions such as penetration of the cervix or the zona pellucida, and motility is an important index for assessing capacitation ability to achieve high fertility rates [33,34]. In the present study, 0.1% and 0.2% κ-carrageenan supplemented into extender resulted in high sperm motility. In agreement with these findings, 0.2–0.8 mg/mL concentrations of κ-carrageenan supplemented into extender have been indicated to induce high motility in rooster sperm after freezing-thawing [22]. Moreover, improved motility of FT spermatozoa following treatment with plant polysaccharides was also reported in boar [35] and bovine [36]. Therefore, cryo-medium containing a low concentration of κ-carrageenan may contribute to the preservation of FT sperm motility.

In the present study, the plasma membrane integrity of FT sperm tended to be higher with 0.1% κ-carrageenan treatment. However, the observed difference was not significant. It has been previously reported that the plasma membrane integrity of spermatozoa decreased significantly after freezing and thawing. The physical and chemical conditions to which spermatozoa are exposed seem to be the most likely cause of this regression [37,38]. The key physical factor that influences cell morphology is the formation of crystal ice outside the cell. The underlying matrix is quickly concentrated as ice forms around the cells, leaving the cells in fluids with a high solute content. Moreover, glycerol, which is commonly used for sperm cryopreservation, has been linked to inducing toxic effects on FT sperm. Indeed, the freezing and thawing process resulted in rapid changes in osmotic pressure which might trigger deformations in the membranous structures [39,40].

Acrosome integrity considers an important indicator of sperm quality. Indeed, the acrosome is a vital organelle that facilitates the passage of the spermatozoa through the zona pellucida of the oocyte prior to fertilization [41]. Our results indicated that 0.5% κ-carrageenan treatment exerts a protective effect on the acrosome integrity of FT sperms. These findings are in line with [22] who reported that a 0.4 mg/mL concentration of κ-carrageenan resulted in a higher percentage of intact sperm acrosome in rooster after freezing-thawing.

Reactive oxygen species (ROS) are generated endogenously and exogenously during sperm cryopreservation. An excessive amount of ROS has been reported to induce a negative impact on sperm quality and stimulates the apoptosis pathway [42,43]. According to our results, κ-carrageenan treatment groups exhibited lower ROS levels than the untreated group. Furthermore, the result indicated that apoptosis was significantly reduced in samples that were frozen in the presence of low κ-carrageenan concentrations as assessed by flow cytometry. A similar finding regarding the reduction of ROS levels after κ-carrageenan treatment of FT rooster semen was indicated by [22]. Moreover, a previous study indicated that carrageenan oligosaccharides and their chemically modified derivatives defend rat thymus lymphocytes from oxidative damage caused by hydrogen peroxide and ultraviolet radiation [44]. Taken together, these findings suggest that treatment of extender with κ-carrageenan might exert a protective role for FT sperm by its antioxidant effect that reduces excess ROS and apoptosis levels. The mechanisms through which κ-carrageenan protects against oxidative stress might be attributed to the sulfate content of this algae polysaccharide [45]. Suppression of H_2_O_2_-induced cellular apoptosis and activation of cellular antioxidant enzymes with marine red algae species have been thoroughly reviewed by Lee et al. [46]. Nevertheless, several studies indicated that superoxide radical scavenging activity positively correlated with the sulfate content of the polysaccharide fractions of seaweeds [45,47,48].

Correspondingly, we analyzed the relative expression of selected genes (BAX, Bcl-2, NOX5, SMOX, and SMS) on control and 0.1% κ-carrageenan treatment. The results indicated an upregulation of the relative expression of Bcl-2. The anti-apoptotic (Bcl-2) gene has a pivotal role in apoptosis regulation [49,50]. On the other hand, the expression of NOX5, SMS, and SMOX genes showed significant down-regulation with κ-carrageenan treatment than the control group. These genes are considered as indicators of oxidative status [51,52,53]. Nevertheless, it has been indicated that excessive levels of ROS are implicated in inducing damage to proteins, nucleic acids, lipids, membranes, and organelles, a process that can trigger apoptosis [54,55]. Taken together, these findings suggest that κ-carrageenan exerts a beneficial effect by preventing apoptosis and inhibiting ROS levels during canine sperm cryopreservation.

## 5. Conclusions

Taken all together, the results of both the motility and apoptosis index confirmed the beneficial effect from the lowest concentrations (0.1% and 0.2%) in the FT sperm quality. Furthermore, gene expression analysis confirmed the aforementioned result as indicated by the downregulation of oxidative state and upregulation of anti-apoptosis-related genes.

## Figures and Tables

**Figure 1 animals-12-00088-f001:**
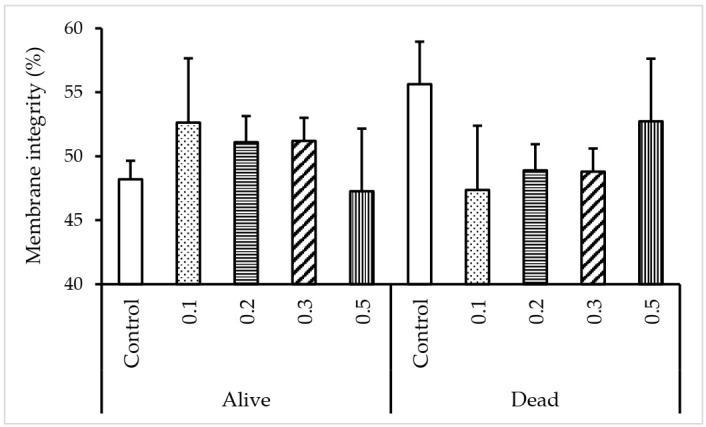
Effect of different κ-carrageenan concentrations in egg yolk-free extender on the plasma membrane integrity of canine sperm following freezing-thawing. FT spermatozoa were stained with Sybr-14/PI and examined under green and red fluorescence. Stained spermatozoa were classified as an intact membrane (Sybr-14+ and considered alive) or damaged membrane spermatozoa (PI+ and considered dead). Results are expressed as the mean ± SE. Error bars indicate the standard error of the mean.

**Figure 2 animals-12-00088-f002:**
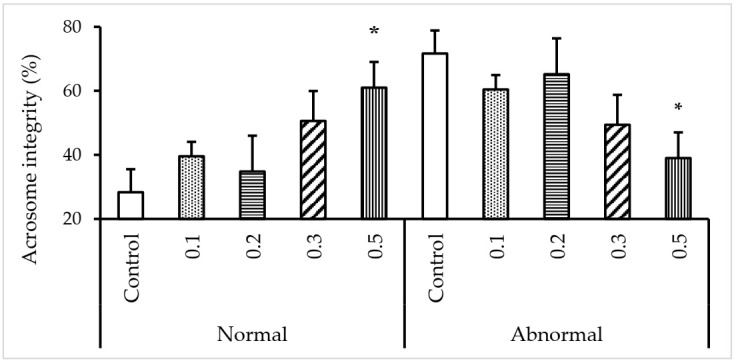
Effect of different κ-carrageenan concentrations in egg yolk-free extender on the acrosome integrity of canine sperm following freezing-thawing. FT spermatozoa were stained with PSA/FITC and examined under green fluorescence. Spermatozoa with intact acrosome were considered normal while those with damaged acrosome were considered abnormal. (*) Above the bar indicate significant differences between groups (*p* < 0.05). Error bars show the standard error of the mean.

**Figure 3 animals-12-00088-f003:**
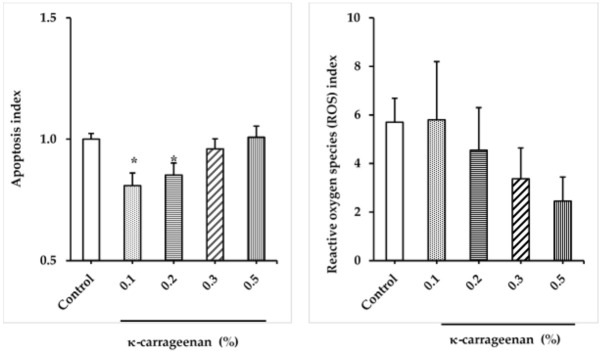
Effect of different κ-carrageenan concentrations in egg yolk-free extender on the apoptosis and the reactive oxygen species (ROS) levels of canine sperm following freezing-thawing. Annexin V-FITC-PI fluorescent staining was used to assess the apoptosis index. The 2′, 7′-dichlorodihydrofluorescein diacetate (H2DCFDA DCF) assay was used to evaluate ROS levels. (*) Above the bar indicate significant differences between groups (*p* < 0.05). Error bars show the standard error of the mean.

**Figure 4 animals-12-00088-f004:**
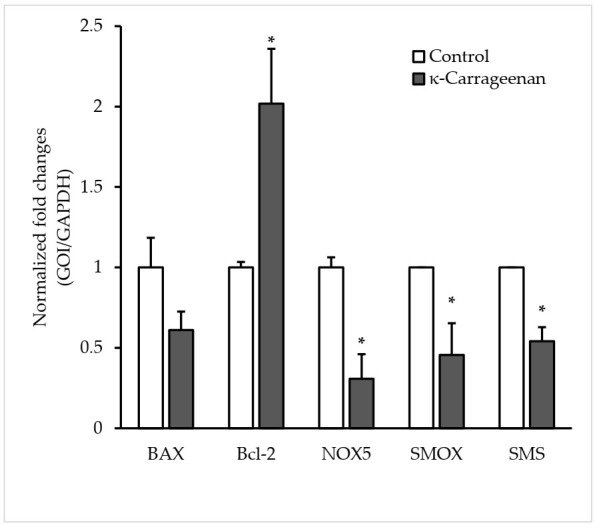
Relative mRNA expression of selected genes following dog sperm cryopreservation using 0.1% κ-carrageenan treatment and (0% control) groups as determined by real-time PCR. GOI: genes of interest. (*) Above the bar indicate statistically significant differences (*p* < 0.05). Error bars show the standard error of the mean.

**Table 1 animals-12-00088-t001:** Primers sequences used for the analysis of relative gene expression.

Gene	Primer Sequences	Product Size (bp)	Accession Number
GAPDH	F: GGA GAA AGC TGC CAA ATA TG R: ACC AGG AAA TGA GCT TGA CA	151	NM_001003142.2
Bcl-2	F: CTC CTG GCT GTC TCT GAA GG R: GTG GCA GGC CTA CTG ACT TC	145	NM_001002949.1
BAX	F: GAC GGC CTC CTC TCC TAC TT R: GGT GAG TGA CGC AGT AAG CA	120	NM_001003011.1
NOX5	F: ACC TGA ACA TCC CCA CCA TC R: TTC AGA CCG GAT GTG TAG CC	101	NM_001103218.1
SMOX	F: AGA AGT GTG ATG ACG AGG CG R: TCG GAA GTA TGG GTT GCT GC	128	XM_855324.3
SMS	F: GTC GCC TGG TTG AGT ATG ACA R: ATG CCA AAT CAC TCT CCG CC	144	XM_005641195.1

F: forward; R: reverse; GAPDH: glyceraldehyde 3-phosphate dehydrogenase; Bcl-2: B-cell lymphoma 2; BAX: Bcl-2-associated X protein; NOX5: NADPH oxidase; SMOX: spermine oxidase; SMS: spermine synthase.

**Table 2 animals-12-00088-t002:** Effect of different κ-carrageenan concentrations in egg yolk-free extender on the motility of canine sperm following freezing-thawing.

Groups	TM (%)	RPM (%)	MPM (%)	IM (%)
Control	40.2 ± 8.3	30.3 ± 4.1	9.9 ± 4.1	59.8 ± 8.3
0.1% κ-carrageenan	63.3 ± 6.8	57.6 ± 6.3 *	5.6 ± 0.4	36.7 ± 6.8
0.2% κ-carrageenan	64.3 ± 0.5 *	56.5 ± 1.4 *	7.8 ± 1.8	35.7 ± 0.5 *
0.3% κ-carrageenan	20.4 ± 0.2	17.7 ± 0.3	2.6 ± 0.5	79.6 ± 0.2
0.5% κ-carrageenan	28.3 ± 9.3	25.3 ± 9.5	3.0 ± 0.2	71.7 ± 9.3

TM: total motility, RPM: rapid progressive motility, MPM: Medium progressive motility, IM: immotility. Values with (*) superscript within the same column are significantly different (*p* < 0.05). The data represent the mean percentages ± standard error.

## Data Availability

The data that support the findings of this study are available upon reasonable request from the corresponding author (Y.J.).

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
