# Peer review of "The Use of κ-Carrageenan in Egg Yolk Free Extender Improves the Efficiency of Canine Semen Cryopreservation"

_animals, 2021, doi:10.3390/ani12010088_

Round 1
Reviewer 1 Report
This is an interesting manuscript related to an atempt of improving canine semen cryopreservation by the addition of κ-Carrageenan to the preservation media. Despite the good results, some corrections are necessary before acceptance. Main of them are listed below.
Abstract: An initial sentence defining what is the κ-Carrageenan would be welcome. Please inform the number of dogs and/or ejaculates used for the experiment. Authors should exclude the sentence related to the improvement of viable sperm in group containing the reagent, since it was not significant. Numeric data would be welcome to ilustrate the results.
Keywords: Try to reorganize your keywords and substitute some of them for words not cited at the title. For example: dog sperm, semen freezing, biobank, etc
Introduction: This section is well written and clearly present justifications for the execution of the experiments. Information related the Carrageenan is really very interesting and sounds scientifically important. Authors should, however, explain why choose the dog as an experimental model. Was the Carrageenan previously tested for the semen of any other species?
Methods: First, the absence of a control group with the diluent added with egg yolk drew attention. While such a group would be interesting, however, its absence does not diminish the merit of the work. In fact, there is a doubt here: the control group was composed for Tris plus glycerol only? without egg yolk?
- Include the CASA settings or reference them.
- The experimental design should be reported at the begining of the section, not at the end. Take care for not being repetitive.
- How many ejaculates were used in a total? This information did not appear anywhere.
Results: Please, do not talk about a possible positive effect of carrageenan on sperm viability. If there were no significant difference, no proved effect exist. Be objective and focus on the results that you really obtained!
Discussion: This section is well written, but authors should present some statements or hypothesis to explain the action mechanism of the carrageenan. Do you know if its chemical composition is known? What would be the active component?
Conclusions: Based on your results, provide an objective conclusion. You do not need to summarize the results again. What concentration of carrageenan do you suggest to be used?
Author Response
"Please see the attachment."

Reviewer 2 Report
Dear authors,
The present study has a wide application in canine semen biotechnologies. However, some issues must be answered before possible acceptance. I suggest that the authors answer the questions point by point.
Keywords: Please do not repeat words already contained in the title
Material and methods
- The experimental design topic must go at the beginning of the methodology, please make this change.
- It is recommended to thaw 0.5mL straws at 37°C for 30 seconds. Why was the protocol used at 38°C for 25 seconds?
- Have samples been diluted for computer assisted sperm analysis? 100 million per mL is a very high concentration, it is recommended to use a maximum of 50 million sperm per mL to obtain an accurate analysis.
- Please include the setup used (frames per second, etc.) and the number of sperm counted for the computer assisted sperm analysis.
- The correct term for "viability" is plasma membrane integrity or membrane permeability. Viability is a feature that goes far beyond membrane integrity.
- How were apoptosis and ROS levels indices calculated?
- Since sperm is a transcriptionally silenced cell, it is difficult to understand why gene expression was performed and how to interpret the results. Authors should clarify the purpose of this analysis.
- For a dose-response curve, linear regression analysis is recommended. I recommend that the authors carry out this analysis to complement the analyzes already carried out. Furthermore, with regression analysis it is possible to find the exact ideal concentration of the treatment.
- The sample number used is relatively low. Did the authors perform the power sample size analysis to determine the sample size? In addition, for ANOVA analysis, a post-hoc curve must be performed.
Results
- Please use the terms membrane/acrosome integrity or damage in the figures.
- For ANOVA analysis, using letters on the bars instead of asterisks facilitates the visualization of significant differences between groups.
Discussion
- I believe the effect of κ-Carrageenan on oxidative homeostasis should be more discussed. What is the target of this compound? Which ROS does it attack? Is the Effect intramitochondrial? Which pathway is involved in the improvement in sperm quality?
- Abortive apoptosis due to failures in spermatogenesis can occur. However, the triggering of apoptosis in mature sperm is still a matter of debate, since sperm do not have transcriptional capacity. Do the authors believe that the improvement in sperm quality was due to prevention of the apoptotic cascade?
Author Response
"Please see the attachment."

Reviewer 3 Report
The current manuscript evaluate the use of k-carrageenan in the cryopreservation media of canine sperm. Motility, viability, acrosome integrity, apoptosis and ROS factors were analyzed in post-freezing-thawing sperm samples.
The results of both motility and apoptosis index confirm the beneficial results from the lowest concentrations (0.1 and 0.2 %) in the post-thawing quality.
There is a major concern with this manuscript:
· How do you explain the sperm mRNA transcription, as long as a long-term dogma establish the absence of transcription in the mature sperm?
The only way to confirm the presence of transcription is to perform new experiments with a transcriptase inhibitor, in order to confirm that the results are not due to the lysis of sperm, or presence of somatic cells in the sperm samples.
Other issues:
· LIN 85. How do you prepare the k-carrageenan? Direct addition into the extender? Or with the use of a vehicle: DMSO, glycerol, etc? Is hydrophilic, lipophilic, etc?
· There is a strong incongruence between results from table 2 (motility) and results from figure 1 (viability). E.g. How it is possible that about 51 % of sperm is alive (group 0.2% k-carrageenan), whereas the motility for this group is 64.3 %. Is there “zombie” sperm, dead and with movement?
· Figure 4. If you solve the believe of transcriptionally silent sperm cell, how do you explain the absence if differences in NOX5, being the mean and the S.E.M: lover than the k-carrageenan SMOX group, the latter with significant differences respect to the control.
Author Response
"Please see the attachment."

Round 2
Reviewer 3 Report
Authors have addressed most of my comments and/or suggestions.